# Enhancing the Mechanical Properties of Historical Masonry Using Fiber-Reinforced Geopolymers

**DOI:** 10.3390/polym15041017

**Published:** 2023-02-17

**Authors:** Ithan Jessemar R. Dollente, Daniel Nichol R. Valerio, Pauline Rose J. Quiatchon, Anabel B. Abulencia, Ma. Beatrice D. Villoria, Lessandro Estelito O. Garciano, Michael Angelo B. Promentilla, Ernesto J. Guades, Jason Maximino C. Ongpeng

**Affiliations:** 1Center for Engineering and Sustainable Development Research, De La Salle University, Manila 1004, Philippines; 2Department of Civil Engineering, De La Salle University, Manila 0922, Philippines; 3Department of Chemical Engineering, De La Salle University, Manila 0922, Philippines; 4Department of Civil Engineering, University of Guam, Mangilao 96923, Guam

**Keywords:** geopolymer, fiber, fiber-reinforcement, alkali-activated, cultural heritage, ambient curing

## Abstract

Current research into the production of sustainable construction materials for retrofitting and strengthening historic structures has been rising, with geopolymer technology being seen as an advantageous alternative to traditional concrete. Fiber reinforcement using this novel cementitious material involves a low embodied carbon footprint while ensuring cohesiveness with local materials. This study aims to develop fly ash-based geopolymers reinforced with six different types of fibers: polyvinyl alcohol, polypropylene, chopped basalt, carbon fiber, and copper-coated stainless steel. The samples are produced by mixing the geopolymer mortar in random distribution and content. Twenty-eight geopolymer mixes are evaluated through compressive strength, split-tensile strength, and modulus of elasticity to determine the fiber mix with the best performance compared with pure geopolymer mortar as a control. Polyvinyl alcohol and copper-coated stainless-steel fiber samples had considerably high mechanical properties and fracture toughness under applied tensile loads. However, the polypropylene fiber source did not perform well and had lower mechanical properties. One-way ANOVA verifies these results. Based on these findings, polyvinyl alcohol and stainless-steel fibers are viable options for fiber reinforcement in historical structures, and further optimization and testing are recommended before application as a reinforcement material in historic structures.

## 1. Introduction

Over the past few years, geopolymer technology has gained much research interest. Geopolymers are a class of materials that have been used for thousands of years, but only in recent decades have they been rediscovered [1]. Dr. Joseph Davidovits reintroduced this technology as he focused on using biopolymers in the field of geomaterials, using low-cost and abundant inorganic raw materials such as sand, fly ash, and volcanic rock to make a durable cementitious material [2,3,4]. Most studies focus on using coal fly ash (CFA), a byproduct of coal-fired power plants that is underutilized and usually ends up in landfills [5,6,7].

Geopolymers have grown in popularity because of their adaptability and potential uses in various industries, especially in construction. They can make concrete, stucco, and ceramic materials [8,9]. They are also lightweight and flexible, have excellent electrical and thermal insulation properties, and resist chemical attacks [10,11,12]. However, one area has not been much examined: cultural heritage preservation [13,14].

The need to preserve and rehabilitate our historical structures has become more vital than ever. Several studies have shown the potential of geopolymers in accomplishing this due to their inherent compatibility with the aggregates and local materials that serve as additives and binding agents [14,15]. Furthermore, because geopolymers are hydrophilic, they are particularly suitable for use in areas with a risk of water seepage into historic structures [16,17,18]. Lastly, geopolymer cement is also ideal for use in areas where constant high humidity makes it difficult to use other cementitious materials [19,20].

Fiber-reinforced concrete has been a significant study scope, and intensive and compelling research has been conducted on reinforcing different fibers with geopolymer [21,22,23,24,25,26]. Fibers are used in composite materials because they enhance the mechanical properties and durability of the composites while being lightweight and ductile [21,27]. Studies on using inorganic, organic, and agro-industrial fibers in reinforcement have shown some significant benefits by imparting specific properties to them. There is also growing literature surrounding ambient curing for fiber-reinforced concrete, as the ambient conditions are flexible for precast and in-situ applications. Shrinkage and cracking are also mitigated in ambient conditions as the matrix is allowed to hydrate slowly and adjust to changes in moisture content [28,29]. Lastly, although compressive strength is lower than in heated conditions, material ductility has been observed to improve in ambient conditions as the slow curing helps the fibers bond to the matrix [30].

Fiber roughness affects tight gripping on the geopolymer matrix, increasing overall material properties [23]. Geopolymerization has been more pronounced for carbon-fiber-reinforced GP concrete, as these fibers increase the nucleation sites for geopolymerization reactions. Moreover, the “bridging effect,” or the intermingling of the fiber with the layers of the matrix and weak surfaces, was empirically observed to increase fracture toughness and decrease crack propagation [10]. These mechanisms were understood better using computerized mesoscale and finite element modeling of the samples. These simulations showed that under loading stress, cracks propagate first in the matrix areas where there are a few concentrations of fibers. Additionally, 3D models show optimal conditions when voids are evenly distributed. This adherence is why one of the biggest challenges in developing fiber-reinforced geopolymer materials is ensuring that the fibers are evenly dispersed throughout the material [31]. Lastly, damage initiation and evolution are mainly influenced by the diameter, length, percent loading, initially observed damages, and spatial distribution of the fibers in the matrix. These studies were mainly conducted for plain mortar, concrete, or fiber-reinforced OPC-based materials, but only recently applied to geopolymer-based materials for a more cost-effective enhancement of properties [32,33,34,35,36].

To further evaluate the adaptability of geopolymer cement as an advantageous alternative to OPC in fiber reinforcement, synthetic and metallic fibers; namely, polyvinyl alcohol (PVA), polypropylene (PP), chopped basalt (CB), carbon fiber (CF), and copper-coated stainless steel (SS) were introduced into the geopolymer matrix, and mechanical properties were assessed to determine the best fiber mix. These results were compared to a plain geopolymer mortar mix.

## 2. Materials and Methods

### 2.1. Raw Materials and Material Characterization

Coal fly ash is a grayish powder sourced from Pozzolanic Philippines, Inc., a coal-fired powerplant located in Calaca, Batangas, Philippines. The XRF analysis of the CFA was performed by the Earth Material Science Laboratory (EMS Lab) in Quezon City, Philippines, and is shown in Table 1. The PVA, CB, CF, and SS fibers were imported from a manufacturer in East China, and the PP fiber was manufactured by Tertex International Philippines Inc., located in Taguig City, Philippines. The CFA and fine aggregates were used as is and needed no further grinding. The alkali activator is made from a viscous water glass solution with a silica modulus of 2.33 and is made from 51.22% Na_2_SiO_3_ solids and NaOH flakes with 98% purity from Formosa Plastics Corporation and imported from Kaohsiung, Taiwan.

### 2.2. Mix Design Formulation

The method used for the experiment design is carried over from the experimental runs of Quiatchon et al. (2021), which is an I-optimal design response surface that used factors to create an optimized formulation [37]. All materials are calculated based on an initial weight of around 3.6 kg of CFA to produce around 15 samples of 50 cm diameter and 100 mm height cylinders and three dog bone samples per run. In this study, the optimal proportions were used based on adjustments to the minimum and maximum runs due to the incorporation of fibers and silica fume. Table 2 shows the factors used in the mixture design of the geopolymer composites.

Different sodium hydroxide (NaOH) solutions were prepared by dissolving appropriate NaOH pellets in tap water and stored in HDPE carboys for 24 h before adding sodium silicate (Na_2_SiO_3_) to produce the alkali activator. After preparing the activator solution with various precursor ratios, it was mixed for six minutes with fly ash, fine aggregates, silica fume, and fibers using a handheld cement mortar mixer. The mortar was transferred to sets of cylinder molds of 50 mm diameter and 100 mm height. The molded samples were then allowed to rest for 48 h before demolding. The samples were then wrapped in plastic and kept in ambient conditions for 28 days. A total of 23 mix designs using the PVA, PP, SS, CB, and CF fibers and 5 PM samples are shown in Table 3.

The mechanical properties of the mixes were determined by testing the compressive, tensile, and flexural strengths of each. The 28-day compressive strength of concrete was among the primary mechanical properties considered when assessing the performance of concrete [38,39,40]. Four to five 50 mm diameter and 100 mm height cylinders were prepared for UCS and split-tensile strength tests. The mechanical strength was equal to the maximum load recorded and was divided into the area of the sample’s surface in contact with the machine. Experiments were carried out using the MATEST E159-01N cement compression machine performed at the Department of Public Works and Highways Bureau of Research and Standards (DPWH-BRS) in Quezon City, Philippines. The machine has a loading rate of 0.2 KN/s for UCS and 0.033 KN/s for tensile strength tests. The experiment was performed under the ASTM C109/C109M method for unconfined compressive strength testing and quality control [41].

### 2.3. Statistical Analysis

To check for significant differences and contributing effects, Jamovi version 2.2, an open-source statistical test and data analysis software, was used to analyze the results [42,43,44]. After inputting data and assumptions, ANOVA and regression analysis were all acquired from the statistical program computation package. The software was also used to acquire density plots, box plots, and statistical tables.

## 3. Results and Discussion

### 3.1. Material Characterization of FRG Samples

The XRD analysis of the plain mortar geopolymer samples in Figure 1 shows the formation of peaks suggestive of a successful geopolymerization process. The most prominent peaks are Anorthite (CaAl_2_Si_2_O_8_) on the 4.04, 3.74, and 3.21 positions. A Quartz (SiO₂) peak was seen at the 3.34 position, a Goethite (α-Fe^3+^O(OH)) peak at the 4.25 position, and a Mullite (3Al_2_O_3_·2SiO_2_) peak at the 2.94 position. This result is verified by the XRF analysis in Table 4, which showed the increased Na_2_O species in the formed geopolymer compared to the fly ash raw material, while the oxide compositions of Ca and Si significantly decreased. It is also notable that the aluminum oxide is not detected in the XRF analysis of both fly ash and formed geopolymer samples, but it is possible that the amounts did not reach the detection limit of the machine but may still be present in both samples.

### 3.2. Compressive Strength

Based on the compressive strength of the cylinder samples, the plain geopolymer mortar had marginally higher compressive strengths than some of the fiber-reinforced geopolymer samples. A generally uniform failure mode is exhibited for all specimens for the compression of samples of fiber-reinforced geopolymer mortar. However, Figure 2a shows the scaling of the surface. The surface scaling shows the expansion or deformation rather than the cracking of geopolymer components and fibers, thus reflecting the weak compressive strength of the PP fiber-reinforced samples. A similar diagonal crack pattern throughout the samples is seen in Figure 2b–e.

A similar situation can be observed from Figure 3 with the CS of the samples with PM samples having a maximum value of 13.6 MPa and a median of 12.6 MPa. SS samples were the highest among the fiber-reinforced geopolymer samples, with a maximum value of 14.8 MPa and a median value of 12.1 MPa, followed by PVA samples with a maximum of 13.0 MPa and a median of 12.0 MPa. CB samples (max = 10.5 MPa, med = 10.1 MPa), CF (max = 10.9 MPa, med = 9.99 MPa). Polypropylene PP had the lowest strengths among all samples (max = 2.20 MPa, med = 1.63 MPa).

The difference between some unreinforced and reinforced samples is evident in other studies due to a combination of curing factors and the porosity of the matrix [45,46,47,48]. The addition of fibers would create additional voids containing unreacted fly ash and form a heterogenous and porous matrix after the geopolymerization process [47]. Non-hydration of FA is more prominent due to the curing process, as the samples in the study were cured at ambient conditions [22]. Another critical factor would be the quality of the samples due to workmanship. The incorporation of fibers also tends to decrease the workability of the matrix and leads to challenges in casting and mold compaction [23]. Moreover, the clumping and agglomeration of fibers in some parts of the matrix would further increase the voids and affect the mechanical strength of the samples [46].

PVA fibers were seen to be superior due to their easy dispersion within the matrix without much aggregation during mixing. Copper-coated stainless steel also worked well due to its inherent nanometric roughness, allowing better contact between the surface and the cementitious matrix [49]. The hydrophilic property of the fibers also leads to short-term gains in compressive strength, as contrasted to plastic fibers being hydrophobic, which leads to short- and long-term shrinkage that affects long-term mechanical strength [23]. In contrast, chopped basalt and carbon fibers were observed to have lower compressive strengths due to the smoothness of the strands. There are not many anchors from the fiber to the geopolymer matrix. These fibers would need additional treatment to induce rough surfaces and positively affect the mechanical strength [17,50,51]. Nonetheless, the main driving force for mechanical strength is the nature of the matrix itself and not the amount or the presence of fibers [52].

### 3.3. Split Tensile Strength

The failure mode of the split-tensile test is shown in Figure 4. The PP fiber-reinforced geopolymer shown in Figure 4a shows similar behavior for its compressive test samples wherein it demonstrates deformation. PP fiber-reinforced geopolymer was observed to be expansive, thus exhibiting less strength capacity in compression and split-tensile tests. A similar failure with a crack generated at the mid-section of the specimens is seen in Figure 4b–e.

The response for the split-tensile strength shown in Figure 5 confirms that the three fiber-reinforced geopolymer samples had greater tensile strength than the plain geopolymer mortar. The greatest tensile strength was for PVA, with about 2.59 MPa as the maximum value and 2.15 MPa as the median value, followed by CB fibers (Max = 1.95 MPa; Med = 1.78 MPa) and SS (Max = 1.83 MPa; Med = 1.69 MPa). The plain mortar samples exhibited a maximum strength of 1.82 MPa and a median of 1.59 MPa. The lowest values are from CF and PP samples, with median values of 1.53 MPa and 0.65 MPa, respectively. A range in values from 0.12 to 0.37 was obtained for the ratio of the split-tensile strength to compressive strength. These results are higher than the observed results of Islam et al., with values of 0.09 to 0.12 for ambient-cured samples of steel fiber integration, and the results of Bhutta et al., with values of 0.11 to 0.12 for PP fiber integration [22,53]. The percentage gain of the split-tensile strength compared with plain mortar is 3.80% for SS, 13.3% for CB, and a maximum percentage gain of 38.0% for PVA fibers applied to the matrix. Similar results were observed from the study of Choi and Yuan, where the increase was about 0.09 to 0.13 in the compressive strength, and the addition of fibers increased the split-tensile strength from 20 to 50% compared to pure mortar [54]. Results of the increase regarding SS, CB, and PVA were evident because an additional load is needed to break the bond of the reinforced matrix before the complete pull-out of the incorporated fibers [47,55,56]. The fibers were seen to add a bridging force to the crack as strain hardening was seen on displacement curves [57].

### 3.4. Responses Based on Different Factors

The density plots in Figure 6 show the response peaks of the compressive and tensile strengths for the different mix designs. Higher peaks were observed when the loading of fibers was maximized, the alkali activator to fly ash ratio was low, the ratio of NaOH to sodium silicate was low, and more aggregates were present in the mixture and are apparent with all fiber-reinforced samples.

### 3.5. Statistical Analysis of Results

As for the statistical analysis of all samples, the number of samples (N), the mean, standard deviation (SD), and standard error (SE) of all runs are compiled in Table 5 and presented as a descriptive box plot in Figure 7. Statistical computations were performed to verify assumptions of normality and variances as the number of samples was not equal. One-way ANOVA was used to determine if the means of each mechanical strength test were statistically equal. The null hypothesis is that all groups (types of fiber) for each strength test have equal means, and the alternative hypothesis is that there is at least one significant difference between the means. Lastly, the level of significance, α, is selected as 0.05.

The *p*-values are presented in Table 6. Based on the *p*-values, it can be observed that there are statistical differences in all groups with a *p*-value of less than 0.001. 

A post hoc test was performed for all these groups to determine the significant differences between the fiber types, as shown in Table 7 and Table 8.

For the compressive and split-tensile strengths, Table 7 and Figure 7a show significant differences in PP samples versus the other fiber-reinforced samples. Pure mortar samples also show a significant difference with three samples (PP, CF, and CB) but are statistically comparable to PVA and SS samples. Lastly, split-tensile strengths shown in Table 8 and Figure 7b indicate the statistical significance of PVA as being higher than all other samples. Meanwhile, PP samples also have significant results, with the inferior results of all the mixes. Lastly, SS, CF, and CB show no statistical difference with pure mortar samples. These results indicate that, in general, there are significant differences in the type of sample fiber loading, and the formulation of the geopolymer matrix would be vital in creating a mix that performs well under specified mechanical conditions. These results also show the superiority of the PVA samples against the other types of fibers for each formulation applied in this study, followed by SS samples. Results also show that PP samples perform poorly compared to this study’s other types of fibers.

### 3.6. Modulus of Elasticity

The modulus of elasticity is a parameter used to forecast how a material will respond to loads. The kind and quantity of fibers employed, together with the characteristics of the matrix material, can each have an impact on the modulus of elasticity of fiber-reinforced concrete. Dog-bone samples were also molded using the best results from the previous mix designs per type of fiber to understand the elastic properties of the fiber-reinforced geopolymer samples to assess this claim. Three to four samples were placed in tension in the longitudinal direction. This test is based on ISO 1920-10:2010 or the determination of the static modulus of elasticity in compression of the hardened concrete samples [58], as shown in Equation (1). The results of the tests are shown in Figure 8 and Figure 9.
(1)$$E_{C}=\frac{\Delta_{\sigma }}{\Delta_{\varepsilon }}=\frac{\sigma_{a}-\sigma_{b}}{\varepsilon_{a}-\varepsilon_{b}}$$
where:

$\sigma_{a}$ is the upper loading stress in N/mm^2^ (MPa); $\sigma_{a}=\frac{F_{c}}{3}$$\sigma_{b}$ is the basic stress in N/mm^2^ (MPa);$\varepsilon_{a}$ is the mean strain under the upper loading stress;$\varepsilon_{b}$ is the mean strain under the basic stress.

**Figure 8 polymers-15-01017-f008:**
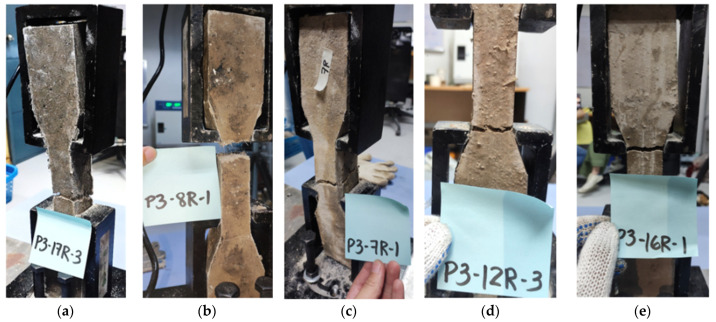
Geopolymer mortar with cracked dog-bone samples of (**a**) PP fibers, (**b**) SS fibers, (**c**) PVA fibers, (**d**) CB fibers, and (**e**) Carbon fibers (CF).

**Figure 9 polymers-15-01017-f009:**
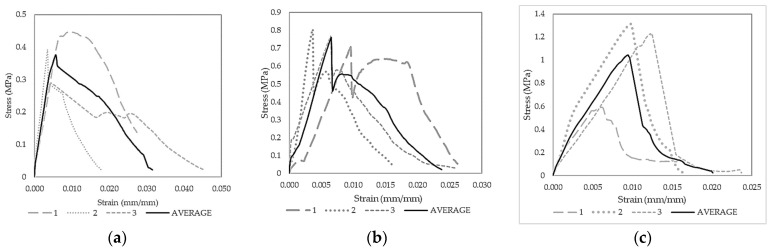
Geopolymer mortar with the stress–strain diagram of (**a**) PP fibers, (**b**) SS fibers, (**c**) PVA fibers, (**d**) CB fibers, (**e**) Carbon fibers (CF), and (**f**) summary of all fiber types.

The failure modes of the dog-bone samples in Figure 8 show macrocracking on the joints. It is observed, however, that the fibers still held the joints of CB and PVA samples after failure. Discrepancies are seen in the stress–strain behavior of the fiber samples. However, these are mainly brought about by adjusting parameters in the mix design of each type of fiber. Similarities of the peak profiles in all samples can be observed and do not vary significantly. The average plots were computed using the equal arc segment method [59]. PP fiber mortars in Figure 9a show an extent on one specimen reaching only about 0.045 strain. The failure mode of specimens was also within the expected failure region, obtaining a modulus of elasticity of 78.54 MPa, and is the lowest among all samples. The stress–strain curve shown in Figure 9b exhibits similar behavior for the SS fiber-reinforced GP samples. Beyond the elastic region, an increase or jump was observed due to the fiber interaction that provides additional strength. The modulus of elasticity was obtained at 125.0 MPa. Failure for all specimens was also consistent within the expected failure region for dog-bone tests. This result shows a good interaction between GP mortar and SS fiber. Figure 9c shows the stress–strain curve of the GP mortar with PVA fiber and the failure mode of each specimen, wherein differences from the failure plane can be observed. One of the PVA samples had stress concentration in the contact surface of the equipment, resulting from cracking initiation. The stress–strain diagram for tension shows the specimens with similar behavior with an approximate modulus of elasticity of 115.4 MPa.

Meanwhile, the behavior throughout the CB fiber, as shown in Figure 9d, exhibits similar peaks and failures after the limit. A 115.32 MPa modulus of elasticity was also obtained. Various failure locations were also observed from the specimens showing brittle failure. Lastly, the CF stress–strain curves show the consistent behavior of specimens with an average modulus of elasticity of 91.45 MPa. As seen in Figure 9e, a straight failure curve was exhibited for all specimens, unlike the Cu-coated steel fiber after the yield point. Lastly, the summary of averages of all fibers is seen in Figure 9f, showing the superior elastic strength of PVA, SS, and CB fibers.

### 3.7. SEM Imaging

Based on the SEM images of the formed fiber-infused samples shown in Figure 10, ample geopolymerization has indeed taken place on the samples where high compressive and tensile strengths were observed, such as with the PVA and SS fibers. For SS samples, there is a cementitious matrix observed under the loose particles, which may have only been due to the area being sampled. This case is different for the CB and CF samples, which have a lot of unreacted fly ash spheres, while gaps were seen on the PP samples where the lowest compressive and tensile strengths were observed. The SEM imaging confirms that these non-hydrated particles and voids did not undergo the geopolymerization process, which has, in turn, affected the mechanical strength of the composite [22].

## 4. Conclusions and Recommendations

This study aims to develop fly ash-based geopolymer (FG) mixes reinforced with various synthetic and metallic fibers. The following results were observed:The plain geopolymer mortar had comparable compressive strengths with PVA and SS samples, followed by CB and CF fiber-reinforced geopolymer samples. PVA fibers had the best performance due to their easy dispersion within the matrix, and copper-coated stainless-steel fibers also performed well due to their nanometric roughness. Due to their smoothness, chopped basalt and carbon fibers had lower compressive strengths, but additional treatment to induce rough surfaces may improve their performance.The PP samples fared the poorest on the compressive test and split-tensile test of samples.The PVA-reinforced fibers also had the highest tensile strength of about 2.18 MPa, followed by CB, SS, and PM fibers. The CF and PP samples had the lowest tensile strengths.The ratio of the split-tensile strength to compressive strength ranged from 0.12 to 0.37. The percentage gain of the split-tensile strength compared to the plain mortar ranged from 3.80% to 38.0%, indicating that an additional load is needed to break the effective bond of the reinforced matrix before the fibers can be pulled out.Statistical analysis revealed significant differences in the all-test means, and multiple comparison tests showed that the PVA samples were superior and the PP samples were inferior to the other fiber types.Dog-bone samples showed that the SS fiber-reinforced samples had good interaction with the geopolymer mortar, but the PP samples had low elastic strain energy and the lowest modulus of elasticity.SEM images showed ample geopolymerization in the PM samples, and the SS samples had a cementitious matrix, leading to high compressive and tensile strengths. The other FRG had many unreacted fly ash spheres and voids that did not undergo geopolymerization, which reduced their mechanical strength.

Based on the results of this study, SS and PVA fibers are viable options for fiber loading and use in historic structures. It is recommended to check further optimization before application and incorporate the aspect ratio as an additional inbound parameter and flexural strength test, adhesion, and bond strength as additional mechanical property tests and response variables. It is also recommended to use other fiber sources to be integrated into the geopolymer mixes, such as rubber and recycled polyethylene fibers. Lastly, as the samples are initially only ambient cured and the intended use is for reinforcement in historic structures, it is also recommended to verify the improvement of the mechanical properties with the incorporation of curing techniques.

## Figures and Tables

**Figure 1 polymers-15-01017-f001:**
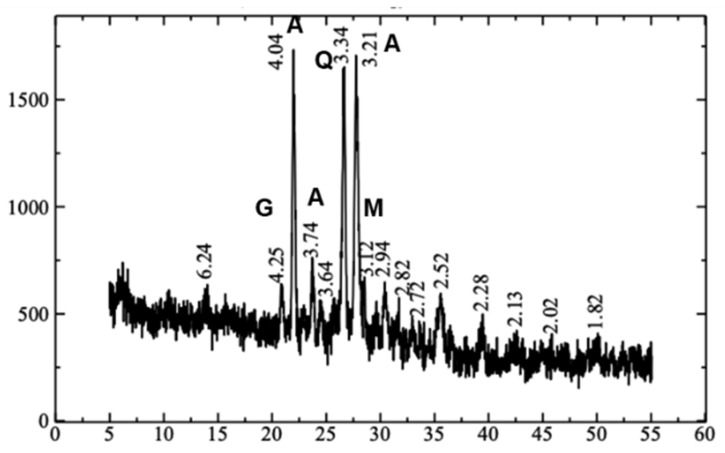
Product characterization of PM samples by XRD.

**Figure 2 polymers-15-01017-f002:**
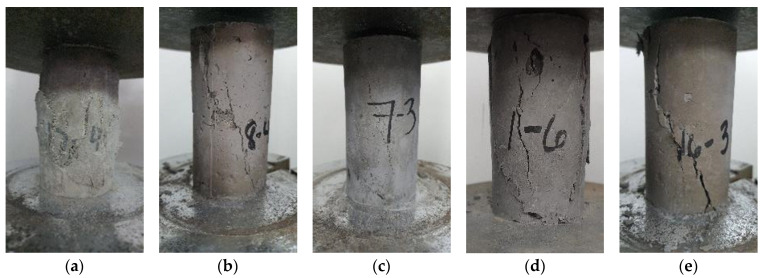
Geopolymer mortar with cracked cylindrical samples of (**a**) PP fibers, (**b**) SS fibers, (**c**) PVA fibers, (**d**) CB fibers, and (**e**) Carbon fibers (CF).

**Figure 3 polymers-15-01017-f003:**
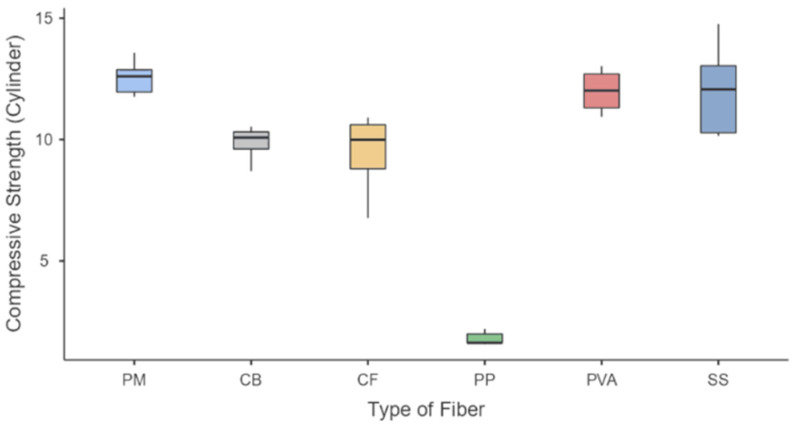
Compressive strength of fiber-reinforced geopolymer.

**Figure 4 polymers-15-01017-f004:**
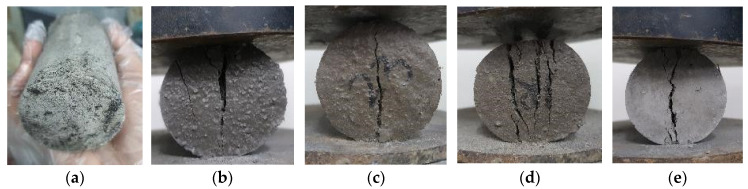
Geopolymer mortar with cracked split-tensile cylindrical samples of (**a**) PP fibers, (**b**) SS fibers, (**c**) PVA fibers, (**d**) CB fibers, and (**e**) Carbon fibers (CF).

**Figure 5 polymers-15-01017-f005:**
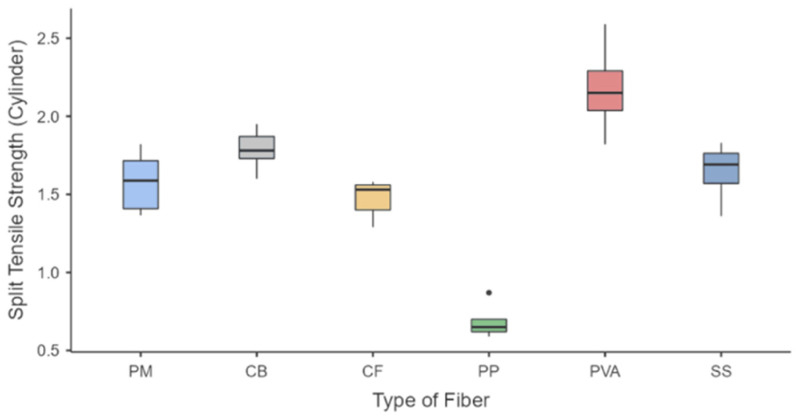
Split Tensile Strength of fiber-reinforced geopolymers.

**Figure 6 polymers-15-01017-f006:**
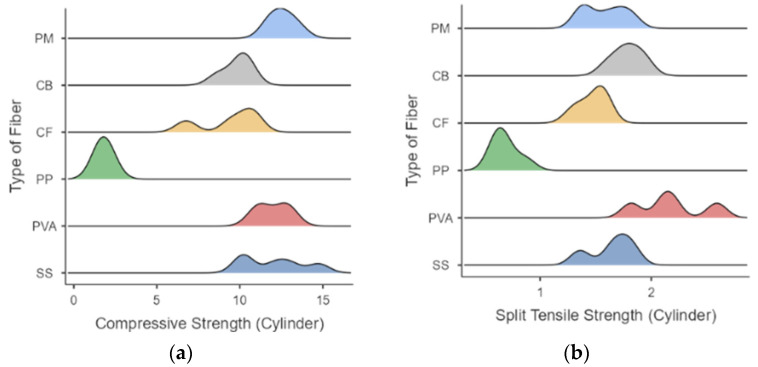
Density plot of responses for (**a**) compressive strength and (**b**) split-tensile strength.

**Figure 7 polymers-15-01017-f007:**
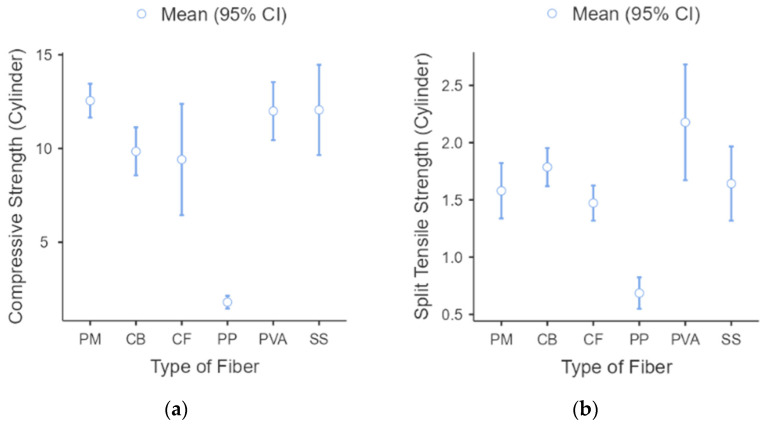
Box plot for FRG samples (**a**) compressive strength and (**b**) split-tensile strength.

**Figure 10 polymers-15-01017-f010:**
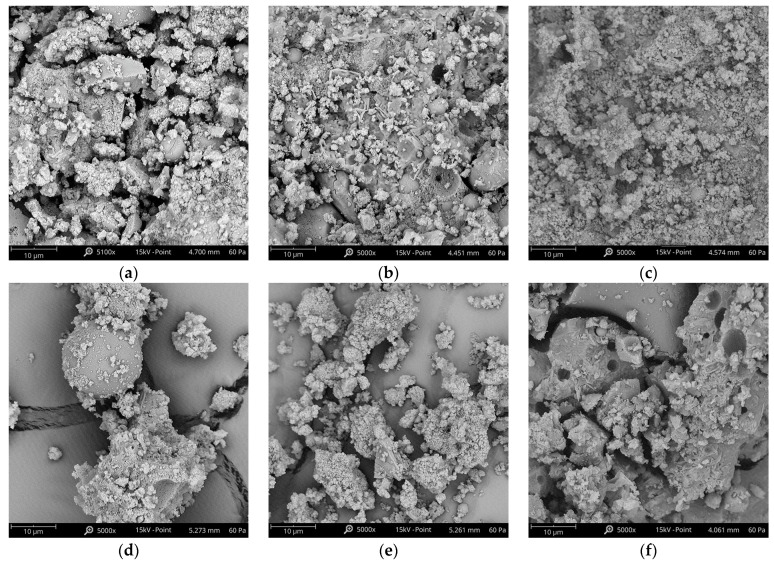
SEM Images of fiber-reinforced GP samples: (**a**) CB fibers, (**b**) CF fibers, (**c**) SS fibers (**d**) PP fibers, (**e**) PVA fibers, and (**f**) Plain Mortar (PM).

**Table 1 polymers-15-01017-t001:** XRF analysis of coal fly ash for this study.

Fly Ash Comp	T1	T2	Fly Ash Comp	T1	T2
MgO	4.740	4.400	K_2_O	1.710	1.490
SO3	1.270	1.170	P_2_O_5_	ND	ND
CaO	6.150	5.660	ZnO	0.022	0.021
SiO_2_	83.400	75.200	MnO	0.110	0.109
Al_2_O_3_	ND	ND	Cl	ND	ND
Fe_2_O_3_	4.140	3.910	Cr_2_O_3_	0.195	0.202
Na_2_O	0.579	0.490	SrO	0.058	0.065
TiO_2_	−0.125	−0.040	Mn_2_O_3_	0.123	0.122

**Table 2 polymers-15-01017-t002:** Parameters of each factor and level for GFRP synthesis [37].

Factor	Name	Minimum	Maximum
1	Alkali Activator/Fly Ash (Precursor) Ratio	0.45	0.50
2	NaOH/Na_2_SiO_3_ Ratio	0.50	0.60
3	Binder/Aggregate Ratio	1.43	1.50
4	Fiber Loading (%)	1.00	2.00
5	Fiber Type	PVA, PP, SS, CB, and CF	

**Table 3 polymers-15-01017-t003:** Experimental runs for PVA, PP, SS, CB, and CF.

Run Code	Alkali Activator-to-Fly Ash (Precursor) Ratio	NaOH to Na_2_SiO_3_ Ratio	Binder to Aggregate Ratio	Silica Fume Loading (%)	Fiber Loading (%)
PM1	0.450	0.600	1.500	10	0.000
PM2	0.450	0.600	1.500	10	0.000
PM3	0.450	0.600	1.500	10	0.000
PM4	0.450	0.600	1.500	10	0.000
PM5	0.450	0.600	1.500	10	0.000
CB1	0.450	0.500	1.425	10	1.075
CB2	0.450	0.500	1.425	10	1.931
PP3	0.450	0.500	1.425	10	1.285
PP4	0.500	0.600	1.500	10	1.000
CF5	0.500	0.600	1.500	10	1.925
CB6	0.450	0.500	1.425	10	1.500
PVA7	0.450	0.500	1.425	10	1.725
SS8	0.450	0.500	1.425	10	1.500
SS9	0.500	0.600	1.500	10	1.065
PVA10	0.500	0.600	1.500	10	2.00
SS11	0.450	0.500	1.425	10	1.500
CB12	0.500	0.600	1.500	10	1.500
CF13	0.500	0.600	1.500	10	1.075
PP14	0.500	0.600	1.500	10	1.715
SS15	0.500	0.600	1.500	10	1.925
CF16	0.450	0.500	1.425	10	1.500
PP17	0.450	0.500	1.425	10	2.000
CF18	0.450	0.500	1.425	10	2.000
PVA19	0.500	0.600	1.500	10	1.285
CB20	0.500	0.600	1.500	10	1.500
PVA21	0.450	0.500	1.425	10	1.000
CF22	0.450	0.500	1.425	10	1.935
PVA23	0.50	0.600	1.500	10	1.075

**Table 4 polymers-15-01017-t004:** Experimental runs for PVA, PP, SS, CB, and CF.

Component	Plain Mortar	
	T1	T2
MgO	2.750	2.770
SO_3_	0.750	0.928
CaO	3.950	3.700
SiO_2_	62.30	56.80
Al_2_O_3_	ND	ND
Fe_2_O_3_	3.990	3.460
Na_2_O	0.663	0.713
TiO_2_	0.101	0.126
K_2_O	0.833	0.841
P_2_O_5_	0.069	ND
ZnO	0.025	0.027
MnO	0.098	0.097
Cl	ND	ND
Cr_2_O_3_	0.137	0.153
SrO	0.040	0.050
Mn_2_O_3_	0.108	0.108

**Table 5 polymers-15-01017-t005:** Group descriptives for analysis of FRG samples.

Test	Type of Fiber	N	Mean	SD	SE
Compressive Strength	PM	5	12.55	0.728	0.325
	CB	4	9.85	0.806	0.403
	CF	4	9.41	1.865	0.932
	PP	5	1.81	0.275	0.123
	PVA	4	11.99	0.974	0.487
	SS	5	12.06	1.937	0.866
Split Tensile Strength	PM	5	1.58	0.195	0.0871
	CB	5	1.786	0.134	0.0599
	CF	5	1.472	0.124	0.0553
	PP	5	0.686	0.111	0.0495
	PVA	4	2.177	0.318	0.1588
	SS	4	1.643	0.204	0.1018

**Table 6 polymers-15-01017-t006:** XRF analysis of coal fly ash for this study.

Test	F	df1	df2	*P*	Remarks
Compressive Strength	51.5	5	21	<0.001	Significant
Split Tensile Strength	32.7	5	22	<0.001	Significant

**Table 7 polymers-15-01017-t007:** Tukey Post-hoc Test for Compressive Strength.

		PM	CB	CF	PP	PVA	SS
PM	Mean difference	—	2.71	3.139	10.75	0.557	0.4936
	*p*-value	—	0.04 *	0.013 *	<0.001 *	0.984	0.988
CB	Mean difference		—	0.433	8.04	−2.148	−2.2119
	*p*-value		—	0.996	<0.001 *	0.189	0.13
CF	Mean difference			—	7.61	−2.582	−2.6453
	*p*-value			—	<0.001 *	0.075	0.047 *
PP	Mean difference				—	−10.189	−10.2524
	*p*-value				—	<0.001 *	<0.001 *
PVA	Mean difference					—	−0.0634
	*p*-value					—	1
SS	Mean difference						—
	*p*-value						—

Note. * *p* < 0.05.

**Table 8 polymers-15-01017-t008:** Tukey Post-hoc Test for Split Tensile Strength.

		PM	CB	CF	PP	PVA	SS
PM	Mean difference	—	−0.206	0.108	0.894	−0.598	−0.0629
	*p*-value	—	0.513	0.938	<0.001 *	0.001 *	0.995
CB	Mean difference		—	0.314	1.1	−0.391	0.1435
	*p*-value		—	0.122	<0.001 *	0.048 *	0.855
CF	Mean difference			—	0.786	−0.705	−0.1705
	*p*-value			—	<0.001 *	<0.001 *	0.745
PP	Mean difference				—	−1.491	−0.9565
	*p*-value				—	<0.001 *	<0.001 *
PVA	Mean difference					—	0.535
	*p*-value					—	0.006 *
SS	Mean difference						—
	*p*-value						—

Note. * *p* < 0.05.

## Data Availability

Not applicable.

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
