# Peer review of "Enhancing the Mechanical Properties of Historical Masonry Using Fiber-Reinforced Geopolymers"

_polymers, 2023, doi:10.3390/polym15041017_

Round 1
Reviewer 1 Report
In the manuscript, fiber-reinforced geopolymers with diferent mixes were investigated with respect to the mechanical properties including compressive strength, split tensile strength and modulus of elasticity. Polyvinyl alcohol (PVA) fibres and copper coated stainless steel (SS) fibres were found to have the optimal performance. XRD analyses and SEM imaging had also been conducted. The manuscript is well presented and is of interest to potential readers. The following points should be properly addressed before a possible recommendation for publication.
(1) The introduction part lacks discussions on the fibre bridging and fibre-mortar interfacial behaviour (see doi.org/10.1016/j.cemconres.2018.01.010; doi.org/10.1016/j.conbuildmat.2022.127013). The two factors have great impacts on the ductility and post-cracking behaviour of fiber-reinforced geopolymers. In addition, recent Computed Tomography (CT) techniques with multiscale capabilities should be mentioned to improve the manuscript, e.g., doi.org/10.1016/j.ijsolstr.2015.05.002 for normal concrete, doi.org/10.1016/j.cemconcomp.2021.104216 for FRC, and doi.org/10.1016/j.jmrt.2022.10.155 for geopolymer composites.
(2) The authors studied the compressive, split tensile and tensile behaviour but only provided the cracked dog-bone samples in Figure 7. The failure pictures of the samples under compression and split tension are suggested to enhance the completeness of the manuscript. The simple sentence "the results of the tests are shown in Figures 7..." does not elucidate the effects of fibre type on the cracking pattern, so an in-depth analysis is required for more insights.
(3) Figure 8, Page 11: the discrepancy between the curves in each figure is significant. The authors should explain this in the manuscript. Also, the reviewer suggests the mean curve in each figure be added for better comparison.
(4) The manuscript needs thorough editing of English and style. For instance, the Abstract, L21: "This study aims to develop fly ash-based geopolymer mixes reinforced with various fibers produced by mixing an alkali activator with fly ash..." The lengthy sentence needs rewriting to improve readability. Section 4, Page 13: Amendments are required to summerise the main conclusions in a point-by-point manner.
Reviewer 2 Report
The authors are congratulated for the document 'Enhancing the Mechanical Properties of Historical Masonry with Fiber-Reinforced Geopolymer: A Case Study on Clay and Adobe Structures', it has a good structure, as well as in the fulfillment of the objectives, references and the practical approach on the analysis and experimental methodology; however, the only pertinent observations that I could comment on are:
Title:
'Enhancing the Mechanical Properties of Historical Masonry with Fiber-Reinforced Geopolymer: A Case Study on Clay and Adobe Structures', the title is very long, it is recommended that it does not exceed 14-16 words and that it generates an attractive and interesting narration.
Abstract:
The Abstract it should be restructured and clearly emphasize the objective of the research, its methodology, and the advantages of the study highlighting its results and conclusions. It is advisable in the abstract to avoid the use of acronyms.
Introduction:
The contribution regarding the use and chronology of polymers is interesting, but it does not contribute anything regarding vernacular architecture in clay, adobe, and stone. For example, take into account that the subject of adobe has a nature, behavior, characterization totally different from clay or stone, and you can not pretend that the same criterion of material serves or is coupled to different realities; in the subject of built heritage, the most complex is the compatibility of mortar, mixture of substitution, this issue is taken very lightly, much more with the international standards of restoration.
It will be necessary to define for which type of heritage architecture the study will be focused and with which the new geopolymer structure is more compatible.
Lines 43 – 58 Revise the structure and wording, the paragraph is very long and dilates the reading, it should be revised and/or divided
Lines 59 – 74 Revise the structure and wording, the paragraph is very long and dilates the reading, it should be revised and/or divided
Materials and Methods
The description of the materials used is clear, but nothing is said about the method used. For similar research, describe the quantitative or semi-quantitative process used by the research, its phases, and other elements (number of samples, weight extracted, prepared weight, texture, imprint, geological nature, etc.)
Results and Discussion
The results of the geopolymer are clear and which of them responded to the expectations of the research, but the A Case Study on Clay and Adobe Structures is not evident, since there is no comparative element "Clay and Adobe”, and any case study could be applied.
Conclusions
Include this section, taking into account that you should focus on each of the findings of the methodology and the results obtained.
Avoid conclusions that are too general and not documented in the text.
References
Universal context references
Round 2
Reviewer 1 Report
The Authors have substantially improved the manuscript, completing all required modifications with in-depth physical interpretations of the results. Therefore, this Reviewer recommends the manuscript for publication in its current form.
Author Response
Dear Reviewer:
Thank you for reviewing our journal submission and for providing insightful criticism. These have contributed to the general success of our journal.
Reviewer 2 Report
The authors are to be congratulated for improving the document both in its structure and its recommendations mentioned before.
Author Response
Dear reviewer:
Thank you very much for your time and effort in helping us improve our manuscript.